# A Modified Green Star Area (MoGSA) and software to assess greenness of reactions in the chemistry laboratories

Fotouh R. Mansour[1,2]*, Mahmoud A. El Hassab[2], Taghreed A. Majrashi[3], Wagdy M. Eldehna[4,5]*

1 Pharmaceutical Analytical Chemistry Department, Faculty of Pharmacy, Tanta University, Tanta, Egypt, 2 Department of Medicinal Chemistry, Faculty of Pharmacy, King Salman International University (KSIU), South Sinai, Egypt, 3 Department of Pharmacognosy, College of Pharmacy, King Khalid University, Asir, Saudi Arabia, 4 Department of Pharmaceutical of Chemistry, Faculty of Pharmacy, Kafrelsheikh University, Kafrelsheikh, Egypt, 5 Department of Pharmaceutical Chemistry, Faculty of Pharmacy, Pharos University in Alexandria, Alexandria, Egypt

* Fotouhrashed@pharm.tanta.edu.eg (FRM); wagdy2000@gmail.com (WME)

**Data Availability Statement:** All relevant data are within the paper.

**Funding:** T.A.M received fund from the Deanship of Research and Graduate Studies at King Khalid

## Abstract

The environmental and health impacts of chemical processes have been a growing concern, leading to the establishment of Green Chemistry principles. Introducing new metrics for the assessment of methods' greenness is crucial to evaluate the exerted efforts to conserve the environment. In this work, we introduce a Modified Green Star Area (MoGSA) and software to assess the greenness of chemical reactions in laboratory settings. MoGSA refines the traditional Green Star Area Index (GSAI) by allowing users to selectively apply specific principles of Green Chemistry based on their relevance to the chemical process being evaluated. This approach addresses the limitations of GSAI, which often lacks clear boundaries between green and non-green practices and does not account for the varying applicability of the 12 Green Chemistry principles across different contexts. Through comparative case studies on catalytic stereoselective reduction of acetophenone, MoGSA demonstrates its utility in providing a more refined and flexible assessment, enhancing both educational and industrial applications of sustainable chemical practices. The software is available as an open source at https://bit.ly/MOGSA.

## Introduction

Awareness of environmental damage and health risks due to chemical industries arose in the mid-20th century [1]. In 1962, Rachel Carson's "Silent Spring" highlighted the dangers of pesticides, sparking the environmental movement [2]. However, it was not until 1990 when Green Chemistry formally emerged, with the U.S. Environmental Protection Agency (EPA) launching the Pollution Prevention Act [3]. Eight years later, Anastas and Warner developed guidelines for designing sustainable chemical reactions and processes [4]. Compliance with these guidelines minimizes the creation of hazardous substances, reducing pollution and waste. It also promotes efficient resource use, which lowers production costs and reduces exposure to

University for funding this work through small group research under grant number (RGP1/121/45). the funders did not play any role in the study design, data collection and analysis, decision to publish, or preparation of the manuscript.

**Competing interests:** The authors have declared that no competing interests exist.

toxic chemicals. This initiative aligned with increasing regulatory demands for safer chemical practices. For these reasons, green chemistry represents a transformative approach in the chemical industry, prioritizing environmental and human health through sustainable practices.

Assessing the greenness of chemical processes is as crucial as setting the guidelines. Metrics provide a quantifiable means to assess the environmental footprint of chemical processes, helping identify areas for improvement [5]. By using these metrics, chemical laboratories can adopt safer, more sustainable, and eco-friendly methodologies [6]. The developed metrics evaluate the toxicity and hazard potential of chemicals, ensuring safer production and usage. It also facilitates comparison between methods, which encourages the development of new, greener technologies and materials [7–9]. Consequently, green chemistry metrics are vital tools for advancing sustainability, safety, and efficiency in the chemical processes [10–13].

There are three types of Green Chemistry Metrics: Mass Metrics, Environmental and Human Health Hazards, and Computational Tools, which utilize software and spreadsheets to calculate and visualize green chemistry metrics, aiding in more precise assessments [14]. Attempts have been exerted to develop comprehensive metrics that integrate these tools and metrics to provide a holistic view of a process's greenness. Among these comprehensive metrics, the Green Star Area Index (GSAI) and EcoScale (ES) are the most common. ES Provides a numerical score from 0 to 100 based on penalty points. To calculate the ES score, penalties are assigned to each element, and the total penalties are subtracted from 100. On the other hand, GSAI holistically evaluates the environmental impact of chemical reactions. In GSAI, the method of greenness is represented as a green star with an area proportional to the degree of compliance with green chemistry principles. It evaluates compliance with the 12 Principles of Green Chemistry, introduced by Anastas and Warner in 1998, which guide the design of environmentally friendly and sustainable chemical processes [4]. In GSAI, each principle is represented by a point on the star, with the extent of compliance determining the length of each point. This metric spans various factors, including human health hazards, environmental risks, resource renewability, and product degradation. The resulting star area visually indicates the overall environmental impact, with a larger green area indicating higher greenness. This makes GSAI a practical tool for chemists, enabling quick and easy comparison of different chemical processes and guiding improvements towards more sustainable practices. It is particularly useful in both educational settings and industrial applications, promoting the broader adoption of green chemistry principles.

However, the current version of GSAI does not set the barriers between green and non-green practices, which compromises decision-making and overall evaluation. Moreover, the 12 principles are not always applicable in all contexts of chemical processes. This makes chemists try to evaluate certain principles of green chemistry while overlooking other less important or practically irrelevant parameters. In this work, we introduce a modified version of Green Star Area (MoGSA), which allows the selection of certain parameters from the 12 principles of green chemistry to apply. The provided open-source software puts the application of this tool at the fingertips of users. MoGSA also makes comparisons between methods easy, rapid, and more reliable.

## Principle of MoGSA

The 12 principles of GC cover different aspects of chemical processes. **Table 1** shows the criteria required to fulfill each principle in GC. These criteria depend on the classification of each substance involved in the reaction and its degradability/renewability. The classification of each substance involved in the reaction is determined based on its potential risks to human health,

**Table 1. Criteria to assess the accomplishment of the principles of green chemistry to construct the green circle (With permission from [29]).**

| Green Chemistry Principle | Criteria for Accomplishment of the Principle |
|---|---|
| P1—Prevention | No waste is produced or has low hazard to human health and the environment (Table 2) |
| P2—Atom Economy | Reactions without excess of reagents (≤10%) and without formation of byproducts (excluding water) |
| P3—Less Hazardous Chemical Synthesis | All substances involved have low hazards to human health and the environment (Table 2) |
| P4—Designing Safer Chemicals | Designed chemicals are non-toxic to humans and the environment (Table 2) |
| P5—Safer Solvents and Auxiliary Substances | Neither solvents nor other auxiliary substances are used, or they have low hazards (Table 2) |
| P6—Increase Energy Efficiency | Environmental pressure and temperature |
| P7—Use Renewable Feedstocks | All raw materials/feedstocks involved are renewable (Table 3) |
| P8—Reduce Derivatives | Derivatizations or similar operations are not used |
| P9—Catalysts | Catalysts are not necessary or have low hazards (Table 2) |
| P10—Design for Degradation | All substances involved are degradable |
| P11—Real-Time Analysis for Pollution Prevention | Continuous monitoring and analysis during the process |
| P12—Safer Chemistry for Accident Prevention | Substances have low hazard for chemical accidents considering health and physical hazards (Table 2) |

the environment, and the likelihood of chemical accidents. This classification is done using a scale ranging from 1 to 3 for the green star or from low to high dangers for the green circle, following the criteria specified in **Table 2**. This table presents the scores awarded to different hazard codes (hazard statements) according to the GHS standards.

The criteria for classifying substances in terms of degradability and renewability are set out in Table 3. The points for each green chemistry principle, as part of constructing the green star, are detailed in **Table 4**.

To illustrate this with an example, the third principle (P3), which represents Less Hazardous Chemical Synthesis, is achieved when all chemicals used in the process have low hazardousness to the environment and health (Table 1). Hazards can be assessed as Physical, Health,

**Table 2. Scores for classifying the hazards of substances using GHS regulations (With permission from [29]).**

| Hazards | Hazard Codes | Score, Green Star | Classification, Green Circle |
|---|---|---|---|
| Physical Hazard Statements | H200, H201, H202, H203, H205, H220, H222, H224, H225, H228 (category 1), H230, H270, H271, H272 (category 2), H240, H241, H242 (Type C and D), H250, H251, H260, H261 (category 2) | 3 | High |
| | H204, H221, H223, H226, H227, H228 (category 2), H229, H231, H272 (category 3), H242 (Type E and F), H252, H261 (category 3), H280, H281, H290 | 2 | Moderate |
| | No indication | 1 | Low |
| Health Hazard Statements | H300, H301, H304, H310, H311, H314, H318, H330, H331, H334, H340, H341, H350, H351, H360, H361, H370, H371, H372, H373 | 3 | High |
| | H302, H305, H312, H315, H317, H319, H332, H335, H336, H362 | 2 | Moderate |
| | No indication | 1 | Low |
| Environmental Hazard Statements | H400, H401, H410, H411, H420 | 3 | High |
| | H402, H412, H413 | 2 | Moderate |
| | No indication | 1 | Low |

**Table 3. Criteria for Classifying Substances Regarding Degradability and Renewability (With permission from [29]).**

| Characteristics | Criteria | Score (S), Green Star | Classification, Green Circle |
|---|---|---|---|
| Degradability | Not degradable and cannot be treated to become degradable to innocuous products | 3 | Not degradable |
| | Not degradable but can be treated to become degradable to innocuous products | 2 | |
| | Degradable and breaks down to innocuous products | 1 | Degradable |
| Renewability | Not renewable | 3 | Not renewable |
| | Renewable | 1 | Renewable |

or Environmental based on the GHS hazard codes in Table 2. Accordingly, chemicals can be categorized as high, moderate, or low in hazardousness, corresponding to S3, S2, and S1, respectively. Consequently, points can be assigned as in Table 4 based on the calculated S score. If all substances are innocuous (S = 1, as shown in Table 3), three points are assigned to the method, and a green radial is given to P3 to indicate the fulfillment of this principle. For substances with a moderate hazard for chemical accidents (S = 2, Table 2), two points are assigned, and a half green/half red radial is used. In cases where substances have a high hazard for chemical accidents (S = 3, Table 2), only one point is assigned, and a red radial represents this level of hazard. If the principle is not applicable, a yellow radial is given to indicate this status. The other principles can be assessed similarly using the same approach.

## Case studies and assessment by MoGSA

The Modified Green Star Area (MoGSA) index evaluates the ecological consequences of chemical processes by taking into account multiple aspects, such as the nature of solvents, reagents, energy usage, and waste production. MoGSA was applied to compare different four different methods for catalytic stereoselective reduction of acetophenone [15]. Two methods applied chemical catalysis using Ru catalysts [16, 17], and two biochemical catalysis using phenylace-taldehyde reductase [18] and *Galactomyces candidus* [19]. The methodology described by Fujii et al. [16] involved the use of a Ruthenium catalyst complexed with a chiral ligand. The catalyst synthesis was performed in a single step using [RuCl$_2$(6-mesitylene)]$_2$ and (1R,2R)-N-(*p*-tolyl-sulfonyl)-1,2-diphenylethylenediamine. The reaction was carried out in a solvent mixture of triethylamine and isopropanol at 28˚C under an air atmosphere. The product was isolated by evaporation of the solvent, achieving a high yield. The Greenness of Fujii et al. revealed high environmental impact with a MoGSA score of as low as 46.67, which indicated an unacceptable degree of sustainability, as shown in **Fig 1**.

The other chemical method [17] utilizes a Ru catalyst that is coordinated with a specific polymeric ligand. The reduction method utilizes NaHCO$_2$ as the reducing agent and water as the solvent. The reaction was performed in an ambient air environment at a moderate temperature of 40˚C. This method was determined by the utilization of non-harmful reagents and conditions, resulting in a higher MoGSA score that demonstrated more alignment with the principles of green chemistry and acceptable sustainability.

On the other hand, the first biocatalytic approach [18] utilized an enzyme derived from a Corynebacterium species. The microorganism was cultivated on a solid substrate and then treated with a cell lysis method to extract the enzyme. Subsequently, this enzyme was employed within a buffer system to facilitate the reduction reaction. The implementation of many sequential procedures for preparation and purification, in conjunction with the utilization of various chemical substances, led to a moderate level of environmental effect, with a MoGSA score of 50, which is comparable with the greenness performance of Itoh et al.

The other biochemical method outlined by Decarlini et al. [19] involved using the fungus strain Galactomyces candidus GZ1 as a biocatalyst for the reduction reaction. The culture

**Table 4. Points to construct the green star (With permission from [29]).**

| Green Chemistry Principle | Criteria | Points |
|---|---|---|
| P1—Prevention | Waste is innocuous (S = 1, Table 3) | 3 |
| | Waste involves moderate hazard to human health and environment (S = 2, Table 3) | 2 |
| | Waste involves high hazard to human health and environment (S = 3, Table 3) | 1 |
| P2—Atom Economy | Reactions without excess reagents (≤10%) and no byproducts | 3 |
| | Reactions without excess reagents (≤10%) and with byproducts | 2 |
| | Reactions with excess reagents (>10%) and no byproducts | 2 |
| | Reactions with excess reagents (>10%) and with byproducts | 1 |
| P3—Less Hazardous Chemical Synthesis | All substances involved are innocuous (S = 1, Table 3) | 3 |
| | Substances involved have moderate hazard (S = 2, Table 3) | 2 |
| | Substances involved have high hazard (S = 3, Table 3) | 1 |
| P4—Designing Safer Chemicals | Designed chemicals are non-toxic to humans and the environment (S = 1, Table 3) | 3 |
| | Designed chemicals have moderate toxicity (S = 2, Table 3) | 2 |
| | Designed chemicals have high toxicity (S = 3, Table 3) | 1 |
| P5—Safer Solvents and Auxiliary Substances | No solvents or auxiliary substances, or they are innocuous (S = 1, Table 2) | 3 |
| | Solvents or auxiliary substances have moderate hazard (S = 2, Table 2) | 2 |
| | Solvents or auxiliary substances have high hazard (S = 3, Table 2) | 1 |
| P6—Increase Energy Efficiency | Room temperature and pressure | 3 |
| | Pressure at room temperature between 0 and 100˚C (cooling/heating required) | 2 |
| | Different pressure or temperature >100˚C or <0˚C | 1 |
| P7—Use Renewable Feedstocks | All raw materials/feedstocks are renewable (S = 1, Table 3) | 3 |
| | At least one raw material/feedstock is renewable (S = 1, Table 3) | 2 |
| | No raw materials/feedstocks are renewable (S = 3, Table 3) | 1 |
| P8—Reduce Derivatives | No derivatizations or similar operations | 3 |
| | Only one derivatization or similar operation | 2 |
| | More than one derivatization or similar operation | 1 |
| P9—Catalysts | No catalysts or innocuous catalysts (S = 1, Table 2) | 3 |
| | Catalysts with moderate hazard (S = 2, Table 2) | 2 |
| | Catalysts with high hazard (S = 3, Table 2) | 1 |
| P10—Design for Degradation | All substances are degradable and break down to innocuous products (S = 1, Table 3) | 3 |
| | Not degradable but can be treated to become innocuous products (S = 2, Table 3) | 2 |
| | Not degradable and cannot be treated to become innocuous products (S = 3, Table 3) | 1 |
| P11—Real-Time Analysis for Pollution Prevention | Continuous monitoring and analysis (S = 1, Table 3) | 3 |
| | Periodic monitoring and analysis (S = 2, Table 3) | 2 |
| | No real-time monitoring or analysis (S = 3, Table 3) | 1 |
| P12—Safer Chemistry for Accident Prevention | Substances have low hazard for chemical accidents (S = 1, Table 2) considering health and physical hazards | 3 |
| | Substances have moderate hazard for chemical accidents (S = 2, Table 2) | 2 |
| | Substances have high hazard for chemical accidents (S = 3, Table 2) | 1 |

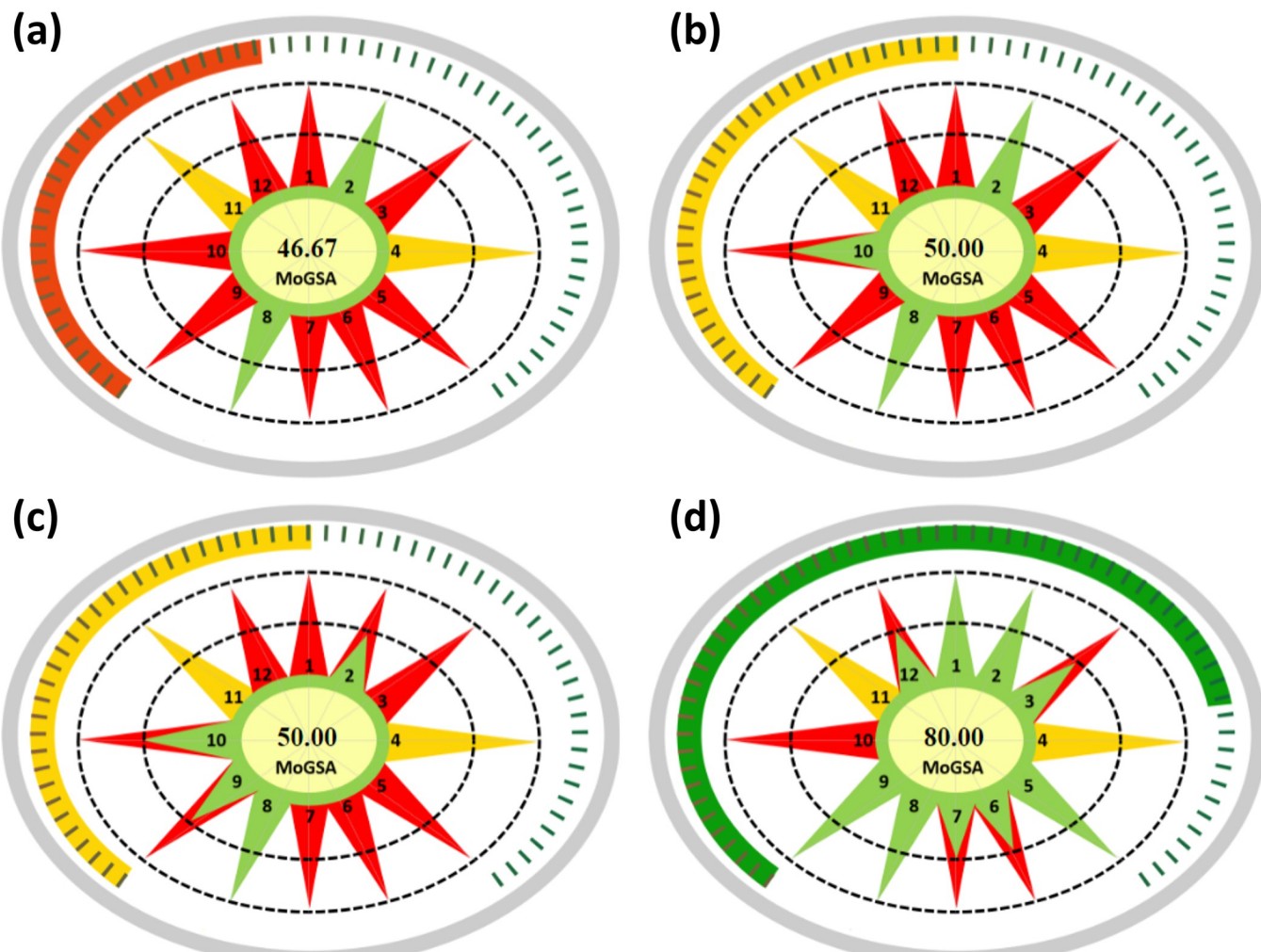

**Fig 1. MoGSA scores for the catalytic stereoselective reduction of acetophenone by Fujii et al [16], Arakawa et al. [17], Itoh et al. [18], and Decarlini et al. [19].**

procedure entailed cultivating the fungus in a buffered medium with a pH of 7 and subsequently using filtering to separate the cells. The implementation of this method yielded a MoGSA score of 80, signifying a significant degree of environmental compatibility. The studied methods demonstrate a correlation between GSA scores and the nature of catalysis, suggesting that both biocatalytic and chemical methods possess distinct advantages and limits. Biocatalytic methods often achieve higher scores on the greenness scale because they utilize renewable resources and operate under gentler reaction conditions. In contrast, chemical methods exhibit greater variability in terms of the reagents and conditions used. Yet, MoGSA scores allow comparisons between different methods, besides enabling the exclusion of selected irrelevant green principles, as needed.

However, MoGSA can also serve educational purposes by choosing the most relevant green principles to implement. For example, Zidny et al. [20] compared the students' assessments of the degree of greenness of three sample preparation techniques, including Soxhlet extraction, microwave-assisted extraction, and steam distillation. The students were required to compare five principles out of the 12 GC principles (P1, P5, P6, P7, P10, and P12) using GSA. Fig 2 presents the same results using the proposed MoGSA tool. Compared with the GSA index,

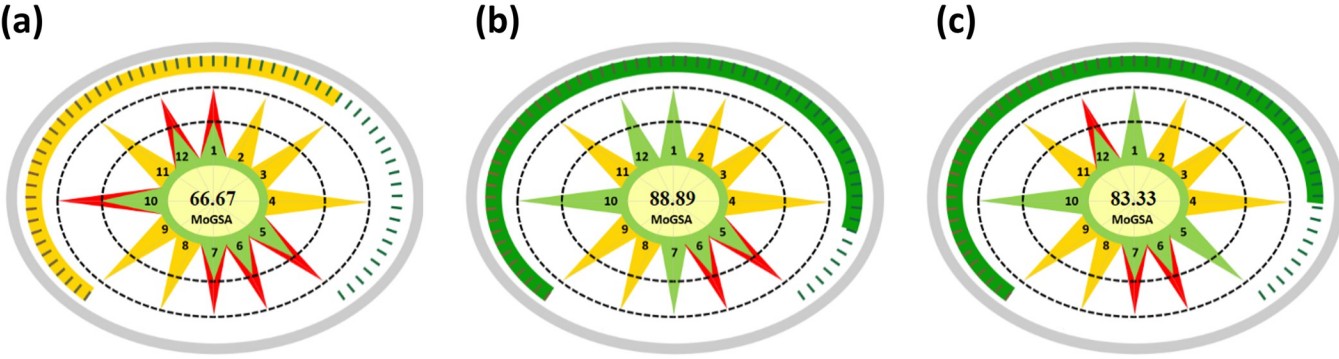

**Fig 2.** MoGSA Scores of Soxhlet extraction (a), microwave-assisted extraction (b), and steam distillation (c).

MoGSA excludes the unselected parameters and automatically calculates the green area based on the compared parameters. The yellow color signifies the parameters excluded from the comparison. Moreover, the total MoGSA score, as indicated by the red, yellow, and green bars, indicates the overall evaluation of the rendered method as either unacceptable, acceptable, or excellent green. This is particularly important in the era of artificial intelligence and machine learning applications [21, 22], as it facilitates prediction and evaluation of greenness in chemical processes.

## Comparison with other greenness assessment metrics

A number of methods have been reported to assess methods' greenness including EcoScale, E-factor (Environmental Factor) [23], Atom Economy [24], ComplexMoGAPI [6] and Life Cycle Assessment (LCA) [25]. The EcoScale evaluates greenness based on penalty points for factors like yield, hazard, cost, and waste but it has limited flexibility for process-specific needs. The E-factor measures the total waste produced per unit of product, offering insight into waste minimization. However, E-factor lacks detailed information on toxicity or energy use. While Atom Economy calculates the efficiency of atom utilization in reactions, it does not consider hazards or environmental impact. ComplexMoGAPI provides a comprehensive scoring system, but it is only suitable for assessing greenness of analytical methods. LCA examines the environmental impact of a product or process over its entire lifecycle, from raw material extraction through disposal. Yet, LCA requires significant data and is often complex to implement. MoGSA metric differentiates itself from existing methods by providing a more adaptable and selective approach for the greenness evaluation. MoGSA allows users to choose principles applicable to the specific process under assessment. It is worth mentioning that atom economy is included in the second principle (P2), while the E-factor, which measures the amount of waste generated per unit weight of the product, is implied in the assessment of the first principle (P1). These factors make MoGSA a more comprehensive and reliable assessment tool. MoGSA's software component also enhances accessibility, which enables straightforward, consistent comparisons and facilitates educational and industrial applications of sustainable practices.

The algorithm used in MoGSA score calculation dynamically processes input data, using JavaScript to calculate and display greenness scores based on user-selected green chemistry principles. It calculates a total score by allowing users to select relevant principles, which are weighted and scored based on reaction-specific input data (e.g., solvent and catalyst types, energy usage). The user-friendly interface features input forms and visual feedback through a color-coded star chart, providing an immediate, customized greenness assessment. This

flexible approach enhances practical application by tailoring the evaluation to each chemical process's unique context. Additionally, the open-source nature of the tool allows easy extension for various chemical reactions, making it useful for educational, laboratory, and industrial applications.

Although these merits highlight how MoGSA can differentiate itself from existing methods of greenness assessment, this does not imply that MoGSA is perfect. A few limitations can be found in the current version of MoGSA. For instance, MoGSA cannot be applied to assess the greenness of methods used in chemical analysis because green analytical chemistry adopts 12 principles different from those of green chemistry. In this case, using more specialized metrics for green analytical chemistry such as MoGAPI [26], ComplexMoGAPI [6], AGREE [27] or AGREEprep [28] is essential. However, this limitation does not diminish the added value of MoGSA or its advantages over other existing methods.

## Conclusion

The Modified Green Star Area (MoGSA) tool provides a significant advancement in the assessment of chemical process sustainability by incorporating flexibility in the application of Green Chemistry principles. This refinement addresses the key limitations of the traditional Green Star Area Index (GSAI), offering a more tailored and relevant evaluation framework. Through detailed case studies, MoGSA has been shown to facilitate more accurate and reliable comparisons between different methods, enhancing decision-making in both educational and industrial settings. By integrating open-source software, MoGSA ensures ease of use and accessibility, promoting broader adoption of green chemistry practices. This innovative approach supports the continuous development and implementation of environmentally friendly and sustainable chemical processes, aligning with the global movement towards reducing environmental and health risks in the chemical industry.

## Author Contributions

**Conceptualization:** Mahmoud A. El Hassab, Wagdy M. Eldehna.

**Data curation:** Fotouh R. Mansour, Mahmoud A. El Hassab, Taghreed A. Majrashi, Wagdy M. Eldehna.

**Formal analysis:** Mahmoud A. El Hassab, Taghreed A. Majrashi.

**Funding acquisition:** Taghreed A. Majrashi.

**Methodology:** Fotouh R. Mansour.

**Resources:** Taghreed A. Majrashi.

**Software:** Fotouh R. Mansour, Mahmoud A. El Hassab, Wagdy M. Eldehna.

**Validation:** Fotouh R. Mansour.

**Visualization:** Wagdy M. Eldehna.

**Writing – original draft:** Fotouh R. Mansour, Mahmoud A. El Hassab.

**Writing – review & editing:** Taghreed A. Majrashi, Wagdy M. Eldehna.

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
