## [Decision Letter · Decision Letter 0]

22 Oct 2024

PONE-D-24-33776A Modified Green Star Area (MoGSA) and Software to Assess Greenness of Reactions in the Chemistry LaboratoriesPLOS ONE

Dear Dr. Eldehna,

Thank you for submitting your manuscript to PLOS ONE. After careful consideration, we feel that it has merit but does not fully meet PLOS ONE’s publication criteria as it currently stands. Therefore, we invite you to submit a revised version of the manuscript that addresses the points raised during the review process.

**ACADEMIC EDITOR: **

Dear Author, Greetings! The Manuscript entitled “A Modified Green Star Area (MoGSA) and Software to Assess Greenness of Reactions in the Chemistry Laboratories” is thoroughly reviewed by experts and we reached to the decision that it needs major revision. Hence, authors are encouraged to revise manuscript as suggested in the reviewer comments. Thanks

We look forward to receiving your revised manuscript.

Kind regards,

Vinaya Satyawan Tari, Post doctoral fellow, (M.Sc., B.Ed., Ph.D.)

Academic Editor

PLOS ONE

Journal Requirements:

“The authors extend their appreciation to the Deanship of Research and Graduate Studies at King Khalid University for funding this work through small group research under grant number (RGP1/121/45).”

“T.A.M received fund from the Deanship of Research and Graduate Studies at King Khalid University for funding this work through small group research under grant number (RGP1/121/45)”

3. We note that your Data Availability Statement is currently as follows: [All relevant data are within the manuscript and its Supporting Information files]

Authors do not need to submit their entire data set if only a portion of the data was used in the reported study

Reviewers' comments:

Reviewer's Responses to Questions

**Comments to the Author**

1. Is the manuscript technically sound, and do the data support the conclusions?

Reviewer #1: Yes

2. Has the statistical analysis been performed appropriately and rigorously? 

Reviewer #1: Yes

3. Have the authors made all data underlying the findings in their manuscript fully available?

Reviewer #1: Yes

4. Is the manuscript presented in an intelligible fashion and written in standard English?

Reviewer #1: Yes

5. Review Comments to the Author

Reviewer #1: The concept of evaluating the "greenness" of chemical reactions has been explored extensively in the literature, with several established metrics already in use. While the paper introduces the Modified Green Star Area (MoGSA), it fails to clearly articulate what differentiates this metric from existing methods. The incremental changes presented seem superficial, and there is no compelling argument as to why MoGSA is a necessary improvement. The lack of significant innovation in the approach makes it difficult to justify the importance of this work.

The rationale behind modifying the original Green Star methodology is not well-explained. The paper lacks sufficient detail about the limitations of the original system and why these necessitate a new version. Without a clear understanding of the weaknesses of previous approaches, the motivation for MoGSA seems forced and unnecessary.

The validation of the MoGSA method and the accompanying software tool is not robust. The authors do not provide enough comparative data to show how MoGSA performs relative to other greenness assessment tools, such as the E-Factor, atom economy, or other lifecycle assessment (LCA) metrics. Additionally, the datasets used for validation are not diverse enough to fully demonstrate the versatility or reliability of MoGSA across different types of chemical reactions.

While the paper highlights the development of a software tool for assessing reaction greenness, it provides very little technical detail about the software itself. There is no clear explanation of the algorithms used, the interface, or how the software processes input data. Moreover, the discussion of the software's usability, scalability, and accuracy is minimal. Without this information, the software's value is unclear, and it seems more like a side note than a substantial contribution.

The theoretical foundation of MoGSA is not well-developed. The modifications appear arbitrary and are not supported by sufficient mathematical or chemical reasoning. A deeper theoretical discussion is needed to explain why these specific changes were made and how they improve upon existing greenness evaluation metrics. Moreover, the paper lacks an in-depth explanation of how MoGSA accounts for the complexities of different chemical processes, such as multi-step reactions or non-linear relationships between input variables.

The practical impact of MoGSA and its software is not clearly demonstrated. There is little discussion of how this metric would be used in real-world laboratory settings, especially when existing greenness assessment tools are already in place. The paper does not provide convincing case studies or examples that show how MoGSA would lead to better decision-making in sustainable chemistry. Without such examples, the practical relevance of this work remains doubtful.

The methodology for calculating the MoGSA score is poorly explained, and there is a lack of clarity regarding how the different parameters are weighted or combined. The description of the methodology is vague, making it difficult for readers to understand the exact process. Additionally, key terms are not well-defined, and the steps of the calculation are not presented in a logical, detailed manner. More transparency in the algorithm would improve the paper’s credibility and allow others to replicate the findings. Update literature with: -Artificial Intelligence and Diagnostic Healthcare Using Computer Vision and Medical Imaging, -Demystifying ChatGPT: An In-depth Survey of OpenAI’s Robust Large Language Models

The paper fails to address the limitations of the proposed MoGSA framework. No greenness metric is perfect, and it is important to discuss the conditions under which MoGSA might perform poorly or offer misleading results. For instance, how does the method handle reactions involving hazardous reagents, large-scale industrial processes, or highly energy-intensive procedures? A more critical assessment of the limitations would add balance and credibility to the discussion.

The writing style is disjointed, and the structure of the paper is hard to follow. Key sections, such as the explanation of the MoGSA methodology and the software tool, are fragmented, making it difficult for readers to grasp the central ideas. Furthermore, the paper contains several grammatical errors and awkward phrasing that distract from the overall message. Improving the clarity and organization of the manuscript would significantly enhance its readability.

The absence of real-world case studies or industrial applications is a major weakness of the paper. Without showing how MoGSA performs in actual laboratory or industrial settings, it is difficult to assess the real impact of the method. Including examples from diverse fields of chemistry would demonstrate the versatility and practical utility of the framework.

6. PLOS authors have the option to publish the peer review history of their article (what does this mean?). If published, this will include your full peer review and any attached files.

Reviewer #1: **Yes: **Dr. Gaurav Dhiman

---

## [Author Response · Author response to Decision Letter 0]

8 Nov 2024

1. The concept of evaluating the "greenness" of chemical reactions has been explored extensively in the literature, with several established metrics already in use. While the paper introduces the Modified Green Star Area (MoGSA), it fails to clearly articulate what differentiates this metric from existing methods. The incremental changes presented seem superficial, and there is no compelling argument as to why MoGSA is a necessary improvement. The lack of significant innovation in the approach makes it difficult to justify the importance of this work.

We thank the reviewer for his time and effort in reviewing our manuscript. A new section has been added to compare the advantages of MoGSA over other existing methods of greenness evaluation (Lines 143-159). 

2. The rationale behind modifying the original Green Star methodology is not well-explained. The paper lacks sufficient detail about the limitations of the original system and why these necessitate a new version. Without a clear understanding of the weaknesses of previous approaches, the motivation for MoGSA seems forced and unnecessary.

The rationale behind modifying the original Green Star methodology has been explained. Details about the limitations of the original system and why these necessitate a new version have been provided (Line 58-66 & Lines 143-159)

3. The validation of the MoGSA method and the accompanying software tool is not robust. The authors do not provide enough comparative data to show how MoGSA performs relative to other greenness assessment tools, such as the E-Factor, atom economy, or other lifecycle assessment (LCA) metrics. Additionally, the datasets used for validation are not diverse enough to fully demonstrate the versatility or reliability of MoGSA across different types of chemical reactions.

Seven case studies were presented to explore the applicability of MoGSA in different chemical processes. Two cases apply chemical catalysis for the stereoselective reduction of acetophenone, two other cases apply biochemical processes, and the remaining three cases compare the degree of greenness of three sample preparation techniques: Soxhlet extraction, microwave-assisted extraction, and steam distillation. These case studies illustrate the versatility, flexibility, and applicability of the proposed MoGSA (Lines 95-142). It is worth mentioning that atom economy is included in the second principle (P2), while the E-factor, which measures the amount of waste generated per unit weight of product, is implied in the assessment of the first principle (P1). These facts make MoGSA a more comprehensive and reliable assessment tool. This has been clarified in the text (Line 158-161)

4. While the paper highlights the development of a software tool for assessing reaction greenness, it provides very little technical detail about the software itself. There is no clear explanation of the algorithms used, the interface, or how the software processes input data. Moreover, the discussion of the software's usability, scalability, and accuracy is minimal. Without this information, the software's value is unclear, and it seems more like a side note than a substantial contribution.

Technical details and explanations have been included in the revised manuscript regarding the MoGSA software's algorithm, and interface. More discussion of the software's usability, applicability, and advantages have been added (Lines 164-172).

5. The theoretical foundation of MoGSA is not well-developed. The modifications appear arbitrary and are not supported by sufficient mathematical or chemical reasoning. A deeper theoretical discussion is needed to explain why these specific changes were made and how they improve upon existing greenness evaluation metrics. Moreover, the paper lacks an in-depth explanation of how MoGSA accounts for the complexities of different chemical processes, such as multi-step reactions or non-linear relationships between input variables.

Explanations why these specific changes were made and how they improve upon existing greenness evaluation metrics have been given (Line 58-66 & Lines 143-159). Explanation of how MoGSA accounts for the complexities of different chemical processes have been given in the case studies (Lines 94-142).

6. The practical impact of MoGSA and its software is not clearly demonstrated. There is little discussion of how this metric would be used in real-world laboratory settings, especially when existing greenness assessment tools are already in place. The paper does not provide convincing case studies or examples that show how MoGSA would lead to better decision-making in sustainable chemistry. Without such examples, the practical relevance of this work remains doubtful.

To demonstrate MoGSA's practical impact, we have included seven case studies that demonstrate its applicability across various chemical processes. Two cases involve chemical catalysis for the stereoselective reduction of acetophenone, two others focus on biochemical processes, and the remaining three evaluate the greenness of sample preparation techniques, including Soxhlet extraction, microwave-assisted extraction, and steam distillation. These examples highlight MoGSA’s adaptability, flexibility, and relevance in real-world laboratory settings. They also emphasize its practical value in advancing sustainable decision-making in chemistry based on the overall evaluation of the methods (Lines 94-142). 

7. The methodology for calculating the MoGSA score is poorly explained, and there is a lack of clarity regarding how the different parameters are weighted or combined. The description of the methodology is vague, making it difficult for readers to understand the exact process. Additionally, key terms are not well-defined, and the steps of the calculation are not presented in a logical, detailed manner. More transparency in the algorithm would improve the paper’s credibility and allow others to replicate the findings. Update literature with: -Artificial Intelligence and Diagnostic Healthcare Using Computer Vision and Medical Imaging, -Demystifying ChatGPT: An In-depth Survey of OpenAI’s Robust Large Language Models.

We thank the reviewer for this comment. The methodology for calculating the MoGSA score has been better explained, and illustrated with an example (Lines 83-91). The literature have been updated with the recommended references (Ref. 21 & Ref. 22)

8. The paper fails to address the limitations of the proposed MoGSA framework. No greenness metric is perfect, and it is important to discuss the conditions under which MoGSA might perform poorly or offer misleading results. For instance, how does the method handle reactions involving hazardous reagents, large-scale industrial processes, or highly energy-intensive procedures? A more critical assessment of the limitations would add balance and credibility to the discussion.

We agree the reviewer that “No greenness metric is perfect “. A more critical assessment of the limitations of MoGSA has been provided (lines 175-182)

9. The writing style is disjointed, and the structure of the paper is hard to follow. Key sections, such as the explanation of the MoGSA methodology and the software tool, are fragmented, making it difficult for readers to grasp the central ideas. Furthermore, the paper contains several grammatical errors and awkward phrasing that distract from the overall message. Improving the clarity and organization of the manuscript would significantly enhance its readability.

We have thoroughly revised the manuscript to enhance clarity and cohesion. The explanation of the MoGSA methodology and software tool has been reorganized into a more logical and continuous flow. We did our best to ensure that key concepts are presented clearly and in a structured manner. We have also carefully proofread the text to address any grammatical errors and awkward phrasing. These revisions have improved the readability and overall quality of the manuscript.

10. The absence of real-world case studies or industrial applications is a major weakness of the paper. Without showing how MoGSA performs in actual laboratory or industrial settings, it is difficult to assess the real impact of the method. Including examples from diverse fields of chemistry would demonstrate the versatility and practical utility of the framework.

Real-world case studies have been added to show how MoGSA performs in actual laboratory conditions and potential industrial settings (Lines 94-142).

---

## [Editor Report · Decision Letter 1]

11 Nov 2024

A Modified Green Star Area (MoGSA) and Software to Assess Greenness of Reactions in the Chemistry Laboratories

PONE-D-24-33776R1

Dear Dr. Eldehna,

We’re pleased to inform you that your manuscript has been judged scientifically suitable for publication and will be formally accepted for publication once it meets all outstanding technical requirements.

Kind regards,

Vinaya Satyawan Tari, Post doctoral fellow, (M.Sc., B.Ed., Ph.D.)

Academic Editor

PLOS ONE

Additional Editor Comments (optional):

Dear Author,

Greetings of the day!

Thank you for carrying out all suggested corrections in the revised version.
---

## [Editor Report · Acceptance letter]

15 Nov 2024

PONE-D-24-33776R1 

PLOS ONE

Dear Dr. Eldehna, 

I'm pleased to inform you that your manuscript has been deemed suitable for publication in PLOS ONE. Congratulations! Your manuscript is now being handed over to our production team.

Kind regards, 

on behalf of

Dr. Vinaya Satyawan Tari 

Academic Editor

PLOS ONE